# Clinical and Instrumental Follow-Up of Childhood Absence Epilepsy (CAE): Exploration of Prognostic Factors

**DOI:** 10.3390/children9101452

**Published:** 2022-09-23

**Authors:** Federico Amianto, Chiara Davico, Federica Bertino, Luca Bartolini, Roberta Vittorini, Martina Vacchetti, Benedetto Vitiello

**Affiliations:** 1Neurosciences Department, Psychiatry Section, Service for Eating Disorders, University of Torino, Via Cherasco 11, 10126 Turin, Italy; 2Department of Pediatrics, Regina Margherita Pediatric Hospital, 10126 Turin, Italy; 3Department of Public Health and Pediatric Sciences, Section of Child and Adolescent Neuropsychiatry, University of Turin, P.zza Polonia 94, 10126 Torino, Italy; 4Hasbro Children’s Hospital, The Warren Alpert Medical School of Brown University, Providence, RI 02912, USA

**Keywords:** childhood absence epilepsy, outcome predictors, seizures, headache, anti-seizure medication

## Abstract

**Background:** Idiopathic generalized epilepsies (IGEs) represent 15–20% of all cases of epilepsy in children. This study explores predictors of long-term outcome in a sample of children with childhood absence epilepsy (CAE). **Methods:** The medical records of patients with CAE treated at a university paediatric hospital between 1995 and 2022 were systematically reviewed. Demographics and relevant clinical data, including electroencephalogram, brain imaging, and treatment outcome were extracted. Outcomes of interest included success in seizure control and seizure freedom after anti-seizure medication (ASM) discontinuation. An analysis of covariance using the diagnostic group as a confounder was performed on putative predictors. **Results:** We included 106 children (age 16.5 ± 6.63 years) with CAE with a mean follow-up of 5 years. Seizure control was achieved in 98.1% (in 56.6% with one ASM). Headache and generalized tonic-clonic seizures (GTCS) were more frequent in children requiring more than one ASM (*p <* 0.001 and *p <* 0.002, respectively). Of 65 who discontinued ASM, 54 (83%) remained seizure-free, while 11 (17%) relapsed (mean relapse time 9 months, range 0–18 months). Relapse was associated with GTCS (*p <* 0.001) and number of ASM (*p <* 0.002). **Conclusions:** A history of headache or of GTCS, along with the cumulative number of ASMs utilized, predicted seizure recurrence upon ASM discontinuation. Withdrawing ASM in patients with these characteristics requires special attention.

## 1. Introduction

Idiopathic generalized epilepsies (IGEs) are a group of epileptic syndromes [1] representing 15–20% of all cases of epilepsy [2]. The International League Against Epilepsy (ILAE) recognizes four syndromes depending on the predominant seizure type, age at onset and EEG findings: epilepsy with generalized tonic-clonic seizures alone (GTCSA), childhood absence epilepsy (CAE), juvenile absence epilepsy (JAE), and juvenile myoclonic epilepsy (JME) [3]. CAE is the most common type, accounting for 10–15% of all childhood epilepsies [3]. Patients with CAE can show absences, myoclonic, and generalized tonic-clonic seizures (GTCS), which can occur alone or in variable combinations.

CAE is generally relatively easy to control with anti-seizure medications (ASM), particularly with ethosuximide (ETX) or valproic acid (VPA), compared to other types of epilepsy [1,4]. Relapse may occur after ASM discontinuation, and, in some cases, seizures continue into adulthood despite best treatment efforts, leading to significant long-term functional consequences and reduced quality of life [5]. Available reports on the long-term outcome of IGEs are heterogeneous for differences in inclusion criteria, outcomes, and duration of follow-up [6]. Remission rates range from 60 to 90% [7,8], and vary by specific syndrome: 92% for GTCSA [9], 56–84% for CAE [6], 37–62% for JAE [6], and 60–90% for JME [9]. Among the negative prognostic factors thus far reported there are: early age of onset [10], psychiatric problems [11], presence of febrile seizures (FS) [7], generalized tonic-clonic seizures (GTCS) [12], and EEG abnormalities [7,8,10]. In CAE, important concerns after the diagnosis are GTCS occurrence [11,13] and cognitive outcome. Many studies, however, are limited by a small sample size and the number of potential predictors examined.

This study aimed to identify predictors of outcome in a sample of patients with CAE, by examining clinical, electroencephalogram (EEG) and treatment characteristics. The two main outcomes of interest were achievement of complete seizure control and maintenance of seizure freedom after ASM discontinuation.

## 2. Materials and Methods

### 2.1. Design and Sample

This was a retrospective study of all the patients with CAE treated at the Division of Child Neurology and Psychiatry of the University of Turin Children’s Hospital Regina Margherita (Turin, Italy), between January 1995 and June 2020. 

Diagnoses were made by paediatric epileptologists, based on clinical presentation, EEG, and brain magnetic resonance imaging (MRI) scans. EEGs were obtained with the international 10–20 system of electrode placement and included hyperventilation and photic stimulation. MRI scans were performed with 1.5 or 3T machines.

The CAE subgroup was defined based on the ILAE Task Force 1989 criteria [14]: (1) onset of typical absences (TAs) before puberty in an otherwise normal child; (2) absence seizures as the predominant seizure type at time of diagnosis; (3) very frequent (several to many per day) absences; and (4) EEG showing bilateral, symmetrical spike-waves, usually 3 Hz, on normal background activity. Patients with TAs under the age of 3 were classified as “Early Onset Absence Epilepsy” (EOAE) and thus excluded from the study, since recent studies have considered it to be a distinct epileptic syndrome [15,16].

A total of 106 records was included according to the following inclusion criteria: (1) diagnosis of CAE based on the ILAE diagnostic criteria; (2) at least one previous EEG performed in our institution showing typical generalized epileptiform discharges of spike-waves or polyspike and waves without any evidence of focality at intake; (3) normal neurological examination; (4) absence of MRI abnormalities possibly triggering seizures at intake; (5) and a follow-up of at least two years.

### 2.2. Outcomes

Primary outcomes were responsiveness to ASM (i.e., full seizure control with no more than 2 ASMs) and seizure-free status after ASM discontinuation.

We further subdivided patients in 3 categories depending on ASM response: 1-Drug Responsive Group (1DR), including children requiring only one drug for seizure control, 2-Drugs Responsive Group (2DR), 3-Resistant Group (3DR), requiring three or more drugs for seizure control. For the characterisation of these three groups, we used the ILAE definition for refractory epilepsy [17].

The definition of remission after treatment and of healing after treatment discontinuation were documented by EEG while awake for at least half an hour with stimulation tests (hyperventilation of 4 to 5 min repeated twice with an interval of one second and photic stimulation) after 3–6 months from withdrawal of drug treatment and, in case of negative EEG, a further observation of 9–6 months.

Patients whose ASM treatment was discontinued were categorized into a Healed-Group, with no seizure recurrence or a Relapsed-Group, in case of seizure recurrence.

### 2.3. Variables Examined as Potential Predictors

Data were extracted from individual medical records by a trained child neurologist. Missing data were integrated, when possible, with patient follow-up, in person or over the telephone. Drug compliance was assessed based on the medical records and ASM serum levels.

The data included: (1) demographic information (gender and age at epilepsy onset); (2) first or second degree family history of epilepsy; (3) prenatal or perinatal injury; (4) FS; (5) seizure types, including status epilepticus (SE); (6) EEG features (focal epileptiform activity and photoparoxysmal response); (7) MRI abnormalities; (8) comorbidities: psychiatric disorders, cognitive impairment, specific learning disability, headache, and sleeping problems (e.g., parasomnia and bruxism; and (9) ASM treatment, including dose, number of ASMs required to manage seizures and cumulative number of ASMs used.

### 2.4. Statistical Analysis

Descriptive statistics were applied. We utilized t-test and one-way ANOVA for continuous variables, and chi-square for categorical variables. We applied a multivariate analysis of the variance (MANOVA) to compare the outcome groups with respect to the clinical variables. In consideration of the explorative nature of the study, we refrained from performing a full Bonferroni correction, and a *p* value of <0.01 was considered statistically significant.

## 3. Results

### 3.1. Sample Characteristics

The cohort consisted of a total of 106 patients affected with CAE, 62 females (58.49%) and 44 males (41.51%), age range 3–35 years. Mean age at epilepsy onset was 5.74 years (SD = 2.48; range: 2–10.5 years), and the mean duration of follow-up was 5.49 years (range: 2–15 years; mean = 5.49; SD = 2.95).

Regarding ASM response, 60 (56.6 %) patients required only one drug to manage seizures (1-Drug Responsive Group), 27 (25.47%) required two drugs (2-Drugs Responsive Group), while 17 patients (16.04%) needed at least 3 drugs (3-Resistant Group). In 2 cases (2%), it was not possible to achieve complete seizure freedom, despite the administration of three or more drugs. Seizure control was achieved with valproate (VPA) monotherapy in 44 (41.5%) patients, with ethosuximide (ETX) monotherapy in 11 (10.4%), and with levetiracetam (LEV) monotherapy in 5 (4.7%). Drug combinations were necessary for 46 (43.4%) patients.

Of the sample, 65 (61.32%) discontinued ASM treatment, and of these 11 (16.92%) resumed ASM due to seizure recurrence. Mean relapse time was 9 months, with a range of 0–18 months.

### 3.2. ASM Response

The 1DR and 2DR showed a less frequent occurrence of headache with respect to 3DR and DU (1.67% in the 1DR, 0% in the 2DR vs. 11.76% in the 3DR, 50% in the DU; *p* < 0.001). Additionally, the GTCS rate was significantly lower in the 1RD and 2 RD with respect to the 3DR group (8.33% in the 1DR, 22.22% in the 2DR vs. 47.06% in the 3DR; *p* < 0.002; Table 1).

### 3.3. Relapse after ASM Discontinuation

The Relapsed Group had higher rate of GTCS (54.5%) compared with the Healed Group (7.4%; *p* < 0.001) and required a higher number of ASMs to manage seizures: 36.4% of the Relapsed Group needed 3 or more drugs versus 5.6% in the Healed Group; *p* < 0.008. All the patients who used 4 or more ASMs belonged to the Relapsed Group (*p* < 0.002; Table 2).

## 4. Discussion

This retrospective study assessed the long-term outcome of CAE with respect to treatment response and investigated predictors of seizure recurrence after discontinuation of ASM treatment. Previous studies reported various and sometimes contradictory results [6,11], possibly due to heterogeneity in sample selection, outcome measures, definition of remission, and duration of follow-up [6]. In this study, we assessed prognosis in terms of both the number of drugs required to manage seizures and occurrence of relapse after ASM withdrawal.

CAE is usually considered to be highly responsive to ASM and to have a good prognosis [18]. In the present study, 98% of the patients achieved control of absence seizures with medications. This is similar to previous reports ranging from 51 to 93% [7,8]. These differences may be due to differences in definitions of seizure control [12]. Some studies define seizure control as seizure freedom for at least 12 months on an unchanged ASM schedule [7], others as seizure freedom after at least 1 year off ASM [19]. Some studies did not specify the duration of seizure-freedom [20,21]. Following ILAE criteria, we considered seizure freedom as at least three times the longest inter-seizure interval [17] with a minimum free-interval of 12 months without using ASM [7].

We found a 17.9% rate of ASM resistance, defined as failure of adequate trials of two ASM schedules to achieve seizure freedom [17]. This finding is consistent with other publications (19–32%) [4,19]. Only a few studies had investigated CAE outcome after ASM discontinuation. In this study recurrence of absence seizures was 16.9%, which is similar to other studies (ranging from 12% to 66%) [22,23].

The number of drugs required to manage seizures was significantly related with the risk of relapses after ASM discontinuation [24,25]. Patients with CAE who respond to the first-line ASM were more likely to reach remission. The poor outcome of non-responders to initial treatment might be either related to a worse initial brain dysfunction or to possible long-term changes and dysfunctions in the brain elicited by the seizures themselves, according to the hypothesis that “seizure begets seizures” [20].

With respect to the therapy, the main predictors of poor response to AED were GTCS and headache. The presence of generalized tonic-clonic seizures (GTCS) was related to both the response to ASM and the occurrence of relapses after AED withdrawal. These findings are in agreement with previous retrospective studies [6,11,12,26,27] but not with a prospective study [21]. In effect GCTS represent a greater brain vulnerability to seizures which may definitely impair drug withdrawal. A recent study [28] suggests that the way in which clinicians manage GTCS significantly affects the prognosis of patients. The first-line treatment for CAE is ethosuximide [29], which is also considered an optimal second-choice AED after failure of an initial treatment [30]. Nevertheless, Cnaan and coworkers [31] focused on valproate prescription after the first-line treatment failure in presence of GCTS: the use of an effective stabilizer as VPA is, may reduce the poorer drug response, while ethosuximide monotherapy is not effective on GCTS. Moreover, the study by Hye Ryun and coworkers [31] indicates that the combined use of VPA and LTG at lower dosage may be more effective than the monotherapy in resistant subjects.

Only headache was associated with an increased risk of relapse and reduced the ASM response rate in the CAE patients. The comorbidity epilepsy/headache is very frequent in CAE, and headaches are more frequent in patients with epilepsy than in the normal population [32]. About 50% of epileptic patients complain of post-critical headache, and at times headache can be the only symptom of an epileptic seizure (ictal epileptic headache) [32]. Both disorders display similar precipitating factors such as lack of sleep, emotional stress, negative feelings [33]. They share several clinical features and have intertwined genetic and molecular underpinnings [34]. Patients with both disorders have longer duration of epilepsy, lower early treatment response, higher incidence of intractable epilepsy, greater need for polytherapy for achieving remission, and in general more problems with seizure control that may lead to ASM adjustment [35]. For instance, Fernandes and coworkers [36] suggested that perampanel may be the best choice for managing comorbid migraine and epilepsy. Moreover, a new onset headache may prompt clinicians to obtain a follow-up EEG to ascertain an undiagnosed epilepsy [37]. Finally, some authors strongly recommend to address comorbid headache, not expecting that the treatment of epilepsy may be necessarily effective on both [38].

Although the usefulness of the EEG for the diagnosis and classification of epilepsy has been widely established [39], its role as a predictor of poor prognosis in IGE remains unclear [6]. The study of Szaflarski and coworkers [8] brought up the presence of focal epileptiform abnormalities as a negative prognostic factor, supporting the theory that the frontal lobes dysfunctions could be the reason for ASM resistance in patients with IGE. More specifically, the study by Dlugos and coworkers [39] evidenced that in CAE longer seizures at baseline indicate more favourable treatment response. Canafoglia and coworkers [40] that the enhanced outflow of frontal oscillations may be helpful to distinguish responders from non-responders. On the contrary, several other publications did not identify the EEG as a useful prognostic predictor [7,10,15,21]. Likewise, in our report no substantial prognostic difference was found between patients with or without focal EEG abnormalities, thus not supporting a relationship between focal dysfunctions and resistance to the ASM treatment. A relationship between status epilepticus and poor prognosis has been described in CAE [11,21]. The lack of significance in the present study is probably due to the small sample size, since only two patients experienced status epilepticus.

## 5. Conclusions

This study further documents the favourable outcome of most cases of CAE and the common use of VPA for achieving seizure control. However, even if only less than 2% of patients were drug resistant and 17% relapsed after ASM withdrawal, a number of clinical characteristics were found to predict treatment response and ultimately persistence of remission upon treatment discontinuation. The prognosis of patients with CAE is related to a clinical history of headache, regardless of its diagnostic definition, and to the occurrence of GTCS after the CAE diagnosis. In fact, these characteristics predicted the need of a higher number of drugs to manage symptoms. Furthermore, GTCS and the number of drugs used to manage absences were predictors of greater risk of relapse after ASM withdrawal. These findings, based on data collected over a relatively long period of observation, can help identify patients at increased risk of relapse and therefore requiring greater attention when considering ASM discontinuation. Moreover, the choice of the ASM combination should be carefully considered for those patients with headache or GCTS to control both CAE and the associated conditions.

### 5.1. What Is Already Known on This Topic

CAE display remission rates from 60 to 90%. Negative prognostic factors are early age of onset, psychiatric problems, febrile seizures, generalized tonic-clonic seizures, and EEG abnormalities.

### 5.2. What This Study Adds

A greater number of ASM required to manage the seizures was related with a higher risk of relapse upon ASM discontinuation. Headache was associated with reduced response to ASM and increased relapse risk upon discontinuation.

### 5.3. How This Study Might Affect Research, Practice or Policy

New-onset headache may prompt EEG to ascertain an undiagnosed epilepsy. Some drugs, e.g., perampanel, are specifically indicated for the treatment of both migraine and epilepsy.

## Figures and Tables

**Table 1 children-09-01452-t001:** **Clinical and EEG characteristics based on the response to AED in patients with CAE**.

Variables	Total n = 106; 100%	1 Drug Responsive Group (1DR) n = 60; 56.6%	2 Drugs Responsive Group (2DR) n = 27; 25.5%	3 Drugs Refractory Group (3DR) n = 19; 17.9%		
*CHI-SQUARE COMPARISON*					* **Χ^2^** *	* **P** *
**Female Gender**	62 (58.0%)	33 (55.0%)	17 (63.0%)	12 (63.2%)	0.694	0.707
**APGAR score at birth < 7**	3 (2.8%)	1 (1.7%)	2 (7.4%)	0 (0%)	2.649	0.266
**Sleep disorders**	7 (6.6%)	6 (10.0%)	1 (3.7%)	0 (0%)	2.834	0.242
**Headache**	4 (3.8%)	1 (1.7%)	0 (0%)	3 (15.8%)	9.347	**0.009**
**Psychiatric disorders**	2 (1.9%)	1 (1.7%)	1 (3.7%)	0 (0%)	0.863	0.650
**Intellectual disability**	7 (6.6%)	3 (5.0%)	2 (7.4%)	2 (10.5%)	2.647	0.619
**Specific Learning Disability**	14 (13.2%)	4 (6.7%)	7 (26.0%)	3 (15.8%)	3.945	0.139
**Febrile Seizures**	9 (8.5%)	5 (8.3%)	1 (3.7%)	3 (15.8%)	2.008	0.366
**Family history of epilepsy**	35 (34.0%)	17 (28.3%)	7 (26.0%)	11 (57.9%)	6.526	0.038
**Generalized Tonic-Clonic Seizures**	19 (17.9%)	5 (8.3%)	6 (22.2%)	8 (42.1%)	11.642	**0.003**
**Status epilepticus**	2 (1.9%)	0 (0%)	1 (3.7%)	1 (5.3%)	2.805	0.246
**Focal abnormalities EEG**	41 (38.7%)	25 (41.7%)	10 (37.0%)	6 (31.6%)	0.660	0.719
**Photo-paroxysmal response**	33 (31.1%)	18 (30.0%)	9 (33.3%)	6 (31.6%)	0.099	0.952
**Abnormal MRI**	17 (16.0%)	10 (16.7%)	4 (14.8%)	3 (15.8%)	0.065	0.968
* **ANOVA COMPARISON** *	*M ± SD*	*M ± SD*	*M ± SD*	*M ± SD*	* **F** *	* **P** *
**Mean age at onset**	5.7 ± 2.5	5.5 ± 2.5	5.9 ± 2.8	6.18 ± 3.2	0.675	0.511

*Note: EEG = electroencephalogram; MRI = magnetic resonance imaging.*

**Table 2 children-09-01452-t002:** **Clinical and EEG characteristics based on the occurrence of relapses in CAE**.

Variables	Total n = 65; 100%	Healed n = 54; 83%	Relapsed n = 11; 16.9%		
* **CHI-SQUARE COMPARISON** *				* **Χ^2^** *	* **P** *
**Female Gender**	39 (60.0%)	31 (57.4%)	8 (72.7%)	0.894	0.344
**Sleep disorders**	5 (7.7%)	5 (8.8%)	0 (0%)	1.103	0.294
**Headache**	2 (3.1%)	1 (1.8%)	1 (9.1%)	1.606	0.205
**Intellectual disability**	3 (4.6%)	2 (3.7%)	1 (9.1%)	0.408	0.523
**Specific Learning Disability**	9 (13.8%)	7 (13.0%)	2 (18.2%)	0.022	0.881
**Febrile Seizures**	5 (7.7%)	4 (7.4%)	1 (9.1%)	0.036	0.849
**Family history of epilepsy**	16 (24.6%)	14 (25.9%)	2 (18.2%)	0.295	0.587
**Generalized Tonic-Clonic Seizures**	10 (15.4%)	4 (7.4%)	6 (54.5%)	15.598	**0.001**
**Status epilepticus**	1 (1.5%)	1 (1.8%)	0 (0%)	0.207	0.649
**Focal abnormalities EEG**	25 (38.5%)	21 (38.9%)	4 (36.4%)	0.025	0.875
**Photoparoxysmal response**	26 (40.0%)	21 (38.9%)	5 (45.4%)	0.164	0.685
**Abnormal MRI**	11 (16.9%)	10 (18.5%)	1 (9.1%)	0.578	0.447
**AED Response**					
1-Drug Responsive	42 (64.6%)	38 (70.4%)	4 (36.4%)	9.728	**0.008**
2-Drugs Responsive	16 (24.6%)	13 (24.1%)	3 (27.3%)		
3-Drug Resistant	7 (10.8%)	3 (5.6%)	4 (36.4%)		
**N° of drugs used during the course of the epilepsy**					
1 drug	40 (61.5%)	37 (68.5%)	3 (27.3%)	17.020	**0.002**
2 drugs	17 (24.1%)	14 (25.9%)	3 (27.3%)		
3 drugs	6 (9.2%)	3 (5.6%)	3 (27.3%)		
4 drugs	1 (1.5%)	0 (0%)	1 (9.1%)		
6 drugs	1 (1.5%)	0 (0%)	1 (9.1%)		
* **t-test COMPARISON** *	*M ± SD*	*M ± SD*	*M ± SD*	** *t* **	** *P* **
**Mean age at onset**	6.1 ± 2.2	5.9 ± 2.2	6.9 ± 2.4	−1.465	0.148

*Note: EEG = electroencephalogram; MRI = magnetic resonance imaging; AED = anti-epileptic drug.*

## Data Availability

Data can be made available by the authors upon qualified request.

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
