# Peer review of "Clinical and Instrumental Follow-Up of Childhood Absence Epilepsy (CAE): Exploration of Prognostic Factors"

_children, 2022, doi:10.3390/children9101452_

Round 1

Reviewer 1 Report

The article is interesting, it presents the problem of epilepsy, which is often a challenge for a doctor who even has extensive experience in its treatment. However, the article requires some corrections and clarifications:

1.What does double hyperventilation mean - please provide a detailed description

2. Male patient data are not included in the tables

3. The discussion discusses in detail the differences between epilepsy and headache, without specifying the type of headache - migraine, tension headaches or other types. I propose to shorten this paragraph because it does not concern the topic directly. Moreover, it is known that headaches occur much more frequently in epilepsy patients than in healthy patients.

4. It is worth paying attention to the treatment, ethosuximide is the drug of choice in CAE, while in patients with GTCS it should not be used as monotherapy because it does not affect this type of seizures.

Author Response

The authors thank the reviewer for the attention paid to the paper and the useful comments. As it concerns the general ratings we improved the references where necessary (citing the names of the authors at line 243 of the first page of discussion). We improved data presentation in the tables as suggested. We amplified and improved conclusions as requested. 

As it concerns the specific points:

  1. we detailed that double hyperventilation means the hyperventilation is repeated twice with 1 second interval;
  2. the tables report the data of both genders, the first line indicates only female data because they are reciprocal with respect to the male and the chi-square indicates the relative risk of being female with respect to being male: i.e. gender relative risk (now we indicated this more clearly);
  3. We reduced the length of the cited paragraph without a differentiation between epilepsy and headache. Moreover we included the suggested sentence.
  4. We underlined in the conclusions the need to pay more attention to treatment, in particular for GTCS

Reviewer 2 Report

The research is very interesting and original. I suggest: review English language, expand the abstract conclusions and expand final conclusions

Author Response

The authors thank the reviewer very much for the attention paid to the paper, the ratings and the useful comments. The English language has been reviewed and some topics have been corrected. The abstract conclusions have been expanded and so the final conclusions. The changes to the text have been tracked with blue fonts.

Round 2

Reviewer 1 Report

all comments have been corrected